

# Estimating intraspecific genetic diversity from community DNA metabarcoding data

Vasco Elbrecht[1,2], Ecaterina Edith Vamos[1], Dirk Steinke[2] and Florian Leese[1,3]

[1] Aquatic Ecosystem Research, University of Duisburg-Essen, Essen, North Rhine-Westphalia, Germany
[2] Centre for Biodiversity Genomics, University of Guelph, Guelph, ON, Canada
[3] Centre for Water and Environmental Research (ZWU) Essen, University of Duisburg-Essen, Essen, North Rhine-Westphalia, Germany

Corresponding author
Vasco Elbrecht,
elbrecht@uoguelph.ca

## ABSTRACT

**Background:** DNA metabarcoding is used to generate species composition data for entire communities. However, sequencing errors in high-throughput sequencing instruments are fairly common, usually requiring reads to be clustered into operational taxonomic units (OTUs), losing information on intraspecific diversity in the process. While Cytochrome c oxidase subunit I (COI) haplotype information is limited in resolving intraspecific diversity it is nevertheless often useful e.g. in a phylogeographic context, helping to formulate hypotheses on taxon distribution and dispersal.

**Methods:** This study combines sequence denoising strategies, normally applied in microbial research, with additional abundance-based filtering to extract haplotype information from freshwater macroinvertebrate metabarcoding datasets. This novel approach was added to the R package "JAMP" and can be applied to COI amplicon datasets. We tested our haplotyping method by sequencing (i) a single-species mock community composed of 31 individuals with 15 different haplotypes spanning three orders of magnitude in biomass and (ii) 18 monitoring samples each amplified with four different primer sets and two PCR replicates.

**Results:** We detected all 15 haplotypes of the single specimens in the mock community with relaxed filtering and denoising settings. However, up to 480 additional unexpected haplotypes remained in both replicates. Rigorous filtering removes most unexpected haplotypes, but also can discard expected haplotypes mainly from the small specimens. In the monitoring samples, the different primer sets detected 177–200 OTUs, each containing an average of 2.40–3.30 haplotypes per OTU. The derived intraspecific diversity data showed population structures that were consistent between replicates and similar between primer pairs but resolution depended on the primer length. A closer look at abundant taxa in the dataset revealed various population genetic patterns, e.g. the stonefly *Taeniopteryx nebulosa* and the caddisfly *Hydropsyche pellucidula* showed a distinct north–south cline with respect to haplotype distribution, while the beetle *Oulimnius tuberculatus* and the isopod *Asellus aquaticus* displayed no clear population pattern but differed in genetic diversity.

**Discussion:** We developed a strategy to infer intraspecific genetic diversity from bulk invertebrate metabarcoding data. It needs to be stressed that at this point this

metabarcoding-informed haplotyping is not capable of capturing the full diversity present in such samples, due to variation in specimen size, primer bias and loss of sequence variants with low abundance. Nevertheless, for a high number of species intraspecific diversity was recovered, identifying potentially isolated populations and taxa for further more detailed phylogeographic investigation. While we are currently lacking large-scale metabarcoding datasets to fully take advantage of our new approach, metabarcoding-informed haplotyping holds great promise for biomonitoring efforts that not only seek information about species diversity but also underlying genetic diversity.

## INTRODUCTION

High-throughput analysis of DNA barcodes retrieved from environmental samples, i.e. DNA metabarcoding, allows for the rapid and standardized assessment of community composition without the need for morpho-taxonomy (*Taberlet et al., 2012a*; *Creer et al., 2016*). This new surge of data enables biodiversity surveys at speeds and scales that were previously inconceivable in ecological and evolutionary studies. While the approach has major strengths and is generally regarded as a game changer for ecological research (*Creer et al., 2016*), it still has limitations such as the fact that sequences are typically clustered into operational taxonomic units (OTUs, Fig. S1) thereby ignoring any intraspecific sequence variation (*Callahan, McMurdie & Holmes, 2017*). However, clustering is often used to reduce the influence of PCR and sequencing errors that can otherwise generate false OTUs (*Edgar, 2013*). The inability to detect sequence variation within OTUs hampers our ability to detect impacts at population level. Simultaneous assessment of inter- and intraspecific diversity, however, represents a leap forward in ecological research and management because haplotype data are direct proxies for spatio-temporal dynamics of populations and both parameters can differ substantially (*Taberlet et al., 2012b*). In particular the assessment of fragmentation (*Weiss & Leese, 2016*) or changes in population size in response to environmental impacts are key areas of basic and applied ecological research (*Sutherland et al., 2012*). For management, this parameter is also important because genetic variation is typically lost long before species or OTUs disappear (*Bálint et al., 2011*). Unfortunately, methods to extract haplotype information from metabarcoding datasets are generally not widely available and thus most studies are based on single-specimen analyses. Some of those are based on denoising algorithms capable of distinguishing between true haplotypes and sequencing noise (*Tikhonov, Leach & Wingreen, 2015*; *Eren et al., 2015*; *Edgar, 2016*; *Callahan et al., 2016*; *Amir et al., 2017*) and have been tested for microbial samples (*Eren et al., 2015*; *Callahan et al., 2016*; *Needham, Sachdeva & Fuhrman, 2017*). *Wares & Pappalardo (2016)* suggested that haplotype information in metazoan datasets can be used to, for instance, improve taxa abundance estimates, which was successfully demonstrated with freshwater fish
fecal samples (*Corse et al., 2017*). Recent studies were also able to infer haplotypes with metabarcoding for single specimens (*Shokralla et al., 2014*), arthropod bulk samples (*Elbrecht & Leese, 2015*; *Pedro et al., 2017*) and environmental water samples (*Sigsgaard et al., 2016*), all highlighting the possibility to extract sequence variant information within OTUs when targeting metazoan taxa.

We here further explored bioinformatics strategies in order to unlock the potential of metabarcoding-based haplotyping of entire and complex metazoan communities. We combined stringent quality filtering of reads with the recently developed *unoise3* denoising strategy (*Edgar, 2016*) and calibrated this approach using a previously characterized single-species mock sample composed of specimens with known haplotypes (*Elbrecht & Leese, 2015*; *Vamos, Elbrecht & Leese, 2017*). Subsequently, we used multi-species metabarcoding data from 18 sample sites that were part of a large-scale governmental freshwater macroinvertebrate biomonitoring program (*Elbrecht et al., 2017*). These were denoised with the developed strategy and we tested the potential to detect intraspecific variation over a broad geographic gradient across multiple taxa.

## MATERIALS AND METHODS

We tested our haplotyping strategy on two available DNA metabarcoding datasets, (1) a single-species mock sample containing 31 specimens with known haplotypes from an earlier population genetics project (*Elbrecht et al., 2014*; *Vamos, Elbrecht & Leese, 2017*) and (2) a multi-species macroinvertebrate community dataset from the Finnish governmental stream monitoring program (*Elbrecht et al., 2017*). Haplotypes were determined by bidirectional Sanger sequencing for the single-species mock samples (*Elbrecht et al., 2014*), while the multi-species sample was metabarcoded on Illumina systems using several primer sets (*Elbrecht & Leese, 2015*, *2017*; *Vamos, Elbrecht & Leese, 2017*). Resulting OTU centroids were assembled into haplotypes as described in *Elbrecht & Leese (2017)*. The samples were sequenced for a region nested within the classical Folmer Cytochrome c oxidase subunit I (COI) region (*Folmer et al., 1994*) with two replicates each. The single-species sample was sequenced using a short primer set amplifying 178 bp (*Vamos, Elbrecht & Leese, 2017*), while the multi-species monitoring samples were amplified using four different primer sets targeting a region of up to 421 bp (*Elbrecht & Leese, 2017*). Paired-end sequencing (250 bp) was performed on Illumina MiSeq and HiSeq systems with great sequencing depth (on average 1.53 million reads per sample, SD = 0.29).

To extract individual haplotypes from the metabarcoding datasets, we used strict quality filtering followed by denoising (unoise3 *Edgar, 2016*, with additional threshold-based filtering steps, see Fig. 1B). The full metabarcoding and haplotyping pipelines are available as part of the "**J**ust **A**nother **M**etabarcoding **P**ipeline" (JAMP) R package (https://github.com/VascoElbrecht/JAMP), which uses Usearch v10.0.240 (*Edgar, 2013*), Vsearch v2.4.3 (*Rognes et al., 2016*) and Cutadapt 1.9 (*Martin, 2011*) for most of the data processing. The advantage of the JAMP wrapper is its modularity and the automated generation of additional summary statistics and extended quality filtering options. All pipeline commands used are also available as Supporting Information (Fig. S2, Scripts S1, JAMP v0.28). In short, pre-processing of reads involved sample demultiplexing,

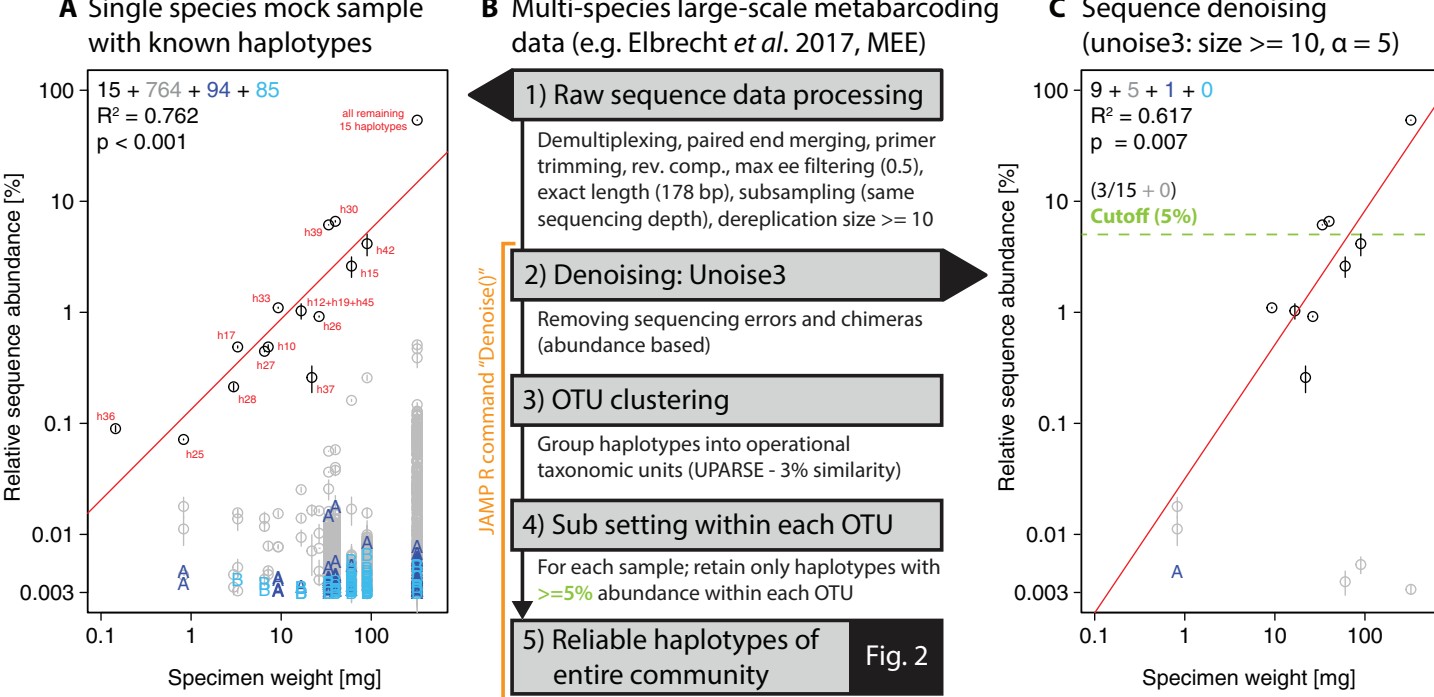

**Figure 1** Overview of DNA metabarcoding data of a single-species mock sample containing specimens with 15 distinct haplotypes amplified with the fwh1 primer set (black circles), with red numbers above each circle showing the original 31 haplotypes using the full 658 bp barcoding region (*Elbrecht & Leese, 2015*; *Vamos, Elbrecht & Leese, 2017*). Detected haplotypes (unexpected ones shown in grey and blue) are plotted against specimen biomass for the processed data (A) and followed by read denoising using unoise3 (C). Denoising was applied to both replicates individually, with a circle if the read was detected in both samples (error bar = SD) and "A" or "B" if the read was found in only one replicate. For processing of the multi-species samples (B, Fig. 2), all samples were pooled and jointly denoised, followed by OTU clustering and read mapping, then followed by discarding of haplotypes below a 5% threshold within each sample.

paired-end merging, primer trimming, generation of reverse complements where needed (to align all reads in the forward direction), maximum expected error (ee) filtering = 0.5 (*Edgar & Flyvbjerg, 2015*), only keeping reads of exact length targeted by the respective primer set and subsampling to 1 and 0.4 million reads, respectively, to generate the same sequencing depth for the single-species and multi-species samples. To further reduce the amount of sequences affected by sequencing errors, we discarded sequences below 10 reads or 0.001% abundance in each sample and applied read denoising with unoise3 after pooling all samples as implemented in Usearch (*Edgar, 2016*) using only reads with ≥10 abundance in each sample after dereplication. Different ee cutoffs and alpha values were tested, with ee = 0.5 and alpha = 5 being used for the final analysis of the 18 monitoring samples. With lower ee values, more low quality sequences were discarded (*Edgar & Flyvbjerg, 2015*). Similarly, lower alpha values led to more strict denoising with unoise3 (*Edgar, 2016*).

For the single-species mock sample, the denoised and quality filtered reads (prior to denoising) were mapped against the expected 15 haplotype sequences using Vsearch (*Rognes et al., 2016*). The unoise3 implementation in the JAMP package adds additional threshold-based filtering after the denoising step, which we used for the Finnish multi-species monitoring samples in order to discard haplotypes with less than 0.01%

abundance in at least one sample and OTUs with less than 0.1% abundance in at least one sample ("Denoise( . . . , minhaplosize = 0.01, OTUmin = 0.1)"). All read mapping steps of denoised data were done with Vsearch. Additionally, within each OTU and sample site, only haplotypes with at least 5% abundance per sample were considered for generating haplotype maps and networks, in order to exclude low abundance OTUs, which can be difficult to separate from PCR artifacts and sequencing errors (withinOTU = 5). The Denoise function also includes presence-based filtering for larger datasets, requiring a specific haplotype or OTU being present in a minimum number of samples (minHaploPresence = 1 or minOTUPresence = 1). However, as we had only 18 sample sites available, this filtering was not applied to the dataset.

## RESULTS

Our approach starts with denoising of quality filtered reads using unoise3 (*Edgar, 2016*) followed by an additional threshold-based filtering step which includes OTU clustering of denoised reads (*Edgar, 2013*) and the removal of low abundant OTUs/haplotypes (see Fig. 1B). We validated this approach by using a single-species mock community of known haplotype composition (*Elbrecht & Leese, 2015*), in which we found 943 unexpected haplotypes above 0.003% abundance with no ee filtering applied (Fig. 1A). Filtering the raw sequence data with different quality thresholds (max ee, *Edgar & Flyvbjerg, 2015*) reduced the number of unexpected haplotypes by only up to 10.22% (Fig. S3). The consistency between the two independent sequencing replicates indicates that a major fraction of the detected haplotypes represent in fact real biological signal (e.g. somatic mutations, numts or heteroplasmy (*Bensasson et al., 2001*; *Shokralla et al., 2014*)), which is difficult to differentiate from PCR and sequencing errors. Even after using different alpha values for the unoise3 algorithm some unexpected sequence variants remained (Fig. S4). An error filtering of max ee = 0.5 in combination with an alpha of 5 was chosen for subsequent analysis (Fig. 1C), as it offers the best trade-off between expected and unexpected haplotypes (nine of 15 expected, six unexpected with low abundance), while retaining 67.08% (SD = 17.69%) of the original sequence data after quality filtering and before denoising.

For the denoising of our multi-species monitoring samples, additional and more conservative filtering steps were introduced to ensure only true sequence variants are included in the analysis (discarding low abundant OTUs and haplotypes below 0.1% and 0.01%, as well as haplotypes below 5% read abundance within each OTU of the respective sample, Fig. 1C green line). Denoising of metabarcoding data from 18 macroinvertebrate samples of the Finnish stream monitoring programme, recovered 177–200 OTUs containing 534–646 haplotypes (on average 2.40–3.30 haplotypes per OTU, SD = 2.13–3.26) for the different primer pairs (Table S1). Most OTUs were only present in a few sample locations, allowing for only limited analyses of intraspecific genetic variation patterns (Fig. S5, see also Fig. S7 in *Elbrecht et al., 2017*). Figure 2 shows some examples of haplotype diversity and geographic distribution for more common and widely distributed taxa in this study. For *Taeniopteryx nebulosa* (Plecoptera) and *Hydropsyche pellucidula* (Trichoptera) we found distinct patterns of latitudinal variation in haplotype composition

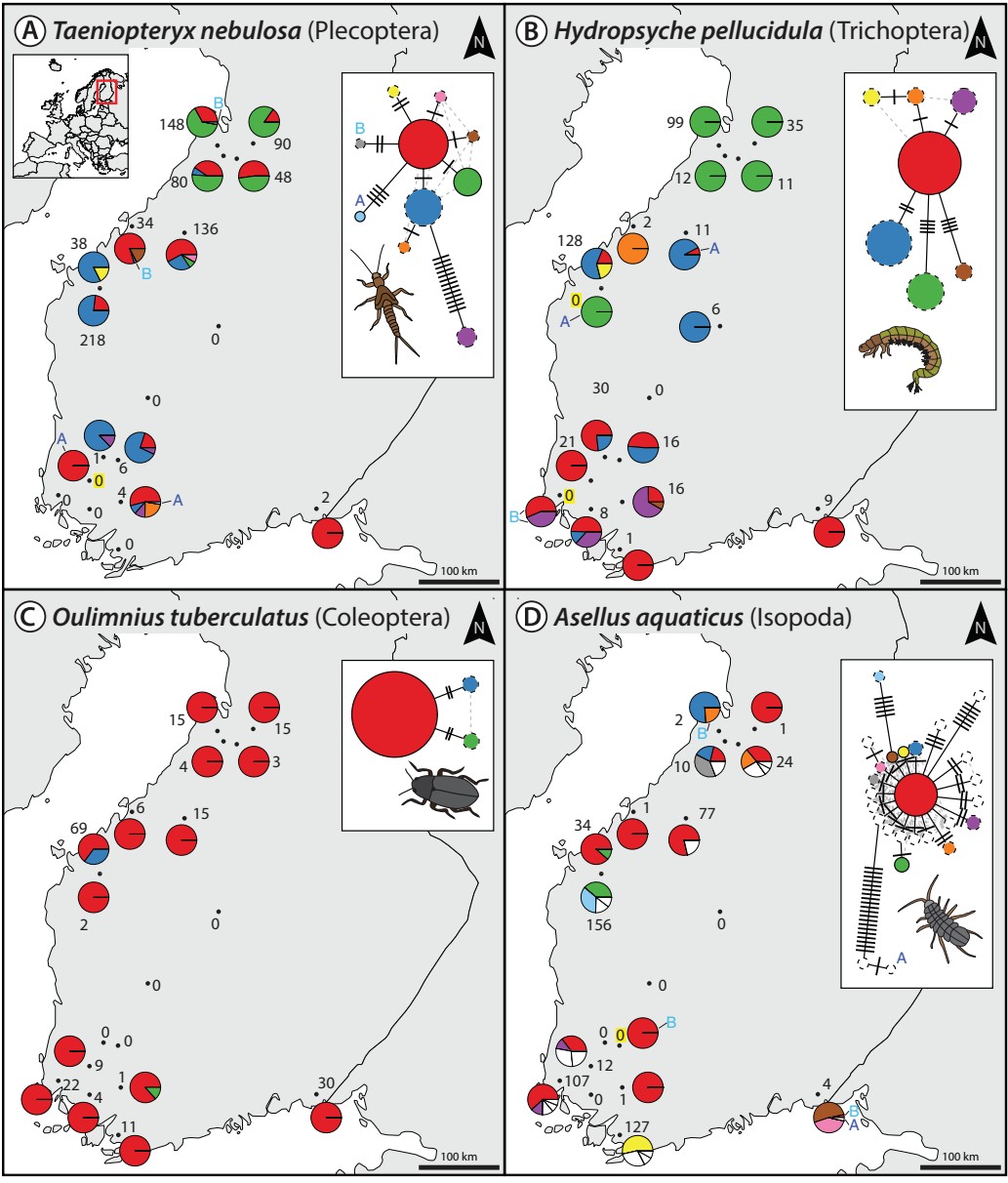

**Figure 2 Haplotype maps and networks extracted from multi-species monitoring metabarcoding datasets amplified with the BF2+BR2 primer set for four abundant macroinvertebrate taxa (A = *Taeniopteryx nebulosa*, B = *Hydropsyche pellucidula*, C = *Oulimnius tuberculatus*, D = *Asellus aquaticus*).** Numbers next to each sampling site indicate sample size of the respective taxa based on morphological identification in a sample (*Elbrecht et al., 2017*). Conflicts between DNA and morphology-based detections are highlighted in yellow. Haplotype frequency composition per site is indicated by pie charts. For *A. aquaticus* only the 10 most common haplotypes are visualised with different colors (remaining ones in white). Each crossline in a network represents one base pair difference between the respective haplotypes. Dashed lines around a circle indicate novel haplotypes that were not available in the BOLD reference database. An "A" or "B" next to a haplotype in the map or network indicates the presence of this haplotype only in one replicate. Shapefile-data© OpenStreetMap contributors, licensed under Creative Commons 2.0 (CC BY-SA).

(Figs. 2A and 2B), while *Oulimnius tuberculatus* (Coleoptera) showed low genetic variation across all primer combinations (Fig. 2C; Fig. S6). *Asellus aquaticus* (Isopoda) on the other hand showed very high genetic diversity and several probably regionally endemic haplotypes (Fig. 2D).

Extracted haplotype patterns between replicates were highly reproducible ($R^2 = 0.751$, SD = 0.242), while at the same time recovering more sequence variants with longer amplicons (Fig. S6). Taxon occurrence for the four taxa analyzed in detail matched morphology based identifications (*Elbrecht et al., 2017*) in most cases (only four false positive detections, Fig. 2). The few inconsistencies between replicates in haplotypes and taxa occurrence are mostly affecting low abundance reads. In the sequence alignments, all four primer sets shared most of the variable positions (Fig. S6).

## DISCUSSION

In this case study, we first developed and demonstrated a bioinformatic strategy to process metabarcoding data using a controlled single-species approach, in order to then extract intraspecific genetic diversity information from complex multi-species metazoan environmental samples. While our multi-species dataset was limited to only 18 sampling sites, and many taxa were not widely distributed (*Elbrecht et al., 2017*), we could still infer COI-based population genetic patterns for some of the abundant and more widespread taxa. Where available, observed population genetic patterns were also consistent with previous studies, e.g. earlier work reported high genetic diversity for *A. aquaticus* (*Sworobowicz et al., 2015*). Other published work, e.g. on *H. pellucidula* (*Múrria et al., 2010*) and *O. tuberculatus* (*Čiampor & Kodada, 2010*), was too limited in sampling size and region for a proper comparison.

Deriving haplotypes from metabarcoding data does not require specialized field or laboratory protocols, as existing data is analyzed. And while our dataset is very limited with just 18 sample sites, there are efforts underway to implement DNA metabarcoding-based monitoring of stream water quality in Europe, potentially generating HTS data for thousands of sample sites every year (*Leese et al., 2016*). Such haplotype data, even though limited in resolution and based only on a single gene marker, could be used to test or derive hypotheses about taxa dispersal and distribution at an unprecedented scale (*Hughes, Schmidt & Finn, 2009*), which would be highly beneficial for basic research but also ecological restoration and management of aquatic ecosystems.

While the detection of haplotypes from bulk samples was demonstrated in this and other studies (*Sigsgaard et al., 2016*; *Corse et al., 2017*; *Pedro et al., 2017*), the limitations of metabarcoding-based haplotyping remain relatively unexplored. Metabarcoding datasets can be affected by primer bias (*Elbrecht & Leese, 2015*), tag switching (*Esling, Lejzerowicz & Pawlowski, 2015*; *Schnell, Bohmann & Gilbert, 2015*), as well as PCR and sequencing errors (*Nakamura et al., 2011*; *Tremblay et al., 2015*). Such issues can lead to artificial haplotypes, which are usually sufficiently different to distinguish them from actual haplotypes in the samples, especially if they are less abundant and thus likely influenced by stochastic effects (*Leray & Knowlton, 2017*). We applied very strict quality filtering in our pipeline, and cautiously discarded all haplotypes below 5% abundance

within an OTU. This is necessary, as low abundant haplotypes cannot be separated from sequencing errors (*Nakamura et al., 2011*; *Tremblay et al., 2015*), somatic mutations (*Shokralla et al., 2014*) and other noise in the data, as we have shown for the single-species mock samples. Strict filtering will remove rare and low abundant haplotypes, but it is necessary to reduce the amount of false positive artificial sequences that result from the currently rather high error rates of HTS instruments. Even with such strict filtering settings, we cannot be fully confident that all false haplotypes were excluded, e.g. as the result of undetected chimeric sequences (*Edgar et al., 2011*) or systematic sequencing errors (*Nakamura et al., 2011*; *Schirmer et al., 2015*; *Schirmer, 2016*) that likely persist across replicates. Approaches relying on the comparison of replicate samples could be an appropriate strategy in particular when working with unicellular organisms (*Lange et al., 2015*). However, for our metazoan communities many variants were found in both replicates (Fig. 1). Macroinvertebrate communities can vary considerably in biomass, which means rare and small specimens will be underrepresented when extracting DNA from bulk samples (*Elbrecht, Peinert & Leese, 2017*). Thus, taxa in the sample are sequenced at different sequencing depth, which likely has an influence on the amount of false haplotypes detected within each OTU. Additionally, differences in specimen biomass can skew the detection of haplotypes, as only those of large specimens will be retained in bioinformatics analysis (haplotypes of small specimens are likely below 5% abundance). Such uncertainties need to be considered when performing population genetic analyses, which are usually done at specimen level, with the exact number of specimens and haplotypes known for each sampling site. It has to be emphasized that at this point metabarcoding-based haplotyping only provides very limited information of genetic diversity and phylogeography of a given taxon. However, interesting patterns emerging from such studies can be subsequently explored by re-collecting taxa of interest and using standard population genetic markers with a higher resolution (e.g. microsatellites or ddRAD *Peterson et al., 2012*). Our study demonstrates the feasibility and potential of metabarcoding data for the investigation of population genetic patterns of entire complex environmental communities. The shortcomings and the level of resolution of this novel approach need to be carefully tested (e.g. by constructing mock samples using synthesized DNA). Additionally, more bioinformatics approaches suited for the analysis of metazoan bulk samples need to be developed, especially with respect to variation in specimen biomass (*Elbrecht, Peinert & Leese, 2017*). Furthermore, most software currently used in this field was developed for microbial samples and should therefore be further tested and benchmarked for its feasibility in studies involving eukaryotes. Despite the clear limitations of this haplotyping approach, we are confident that it will be useful in future large-scale studies of genetic diversity. While metabarcoding studies will remain affected by sequencing errors (potentially leading to false haplotypes), we expect that most of these issues can be mitigated by increasing the number of sampling sites to several hundred or even thousands. For large-scale efforts such as routine monitoring using metabarcoding (*Baird & Hajibabaei, 2012*; *Gibson et al., 2015*; *Elbrecht et al., 2017*), this might soon become a feasible option if not standard. Additionally, references databases should be further completed and extended to cover a large geographic range in

order to assign species names and ground truth the detected haplotypes (*Carew et al., 2017*; *Curry et al., 2018*).

## CONCLUSION

Our study demonstrates that haplotypes can be extracted from complex metazoan metabarcoding datasets. This proof of concept work already shows emerging population genetic patterns for a few species, but more large-scale validation studies are needed to explore the limitations and the potential of metabarcoding-based haplotyping. While some shortcomings such as occasional false positive detections and loss of rare and small taxa for such complex communities are difficult to overcome on a per sample bases, these might be partly offset by studying comparative patterns of intraspecific variation across many taxa and sites. As metabarcoding becomes more accessible and larger DNA-based biodiversity assessment and monitoring initiatives emerge, sampling and extracting haplotypes from hundreds of sites might become a feasible path of future research.

## ACKNOWLEDGEMENTS

We would like to thank members of the LeeseLab for helpful discussions.

### Funding

This study is part of the European Cooperation in Science and Technology (COST) Action DNAqua-Net (CA15219). Dirk Steinke and Vasco Elbrecht were supported by the Canada First Research Excellence Fund for the Food from Thought initiative. Ecaterina Edith Vamos was supported by a grant of the Bodnarescu Foundation (Deutsches Stiftungszentrum). The funders had no role in study design, data collection and analysis, decision to publish, or preparation of the manuscript.

### Grant Disclosures

The following grant information was disclosed by the authors:
European Cooperation in Science and Technology (COST) Action DNAqua-Net: CA15219.
Canada First Research Excellence Fund.
Bodnarescu Foundation.

### Competing Interests

The authors declare that they have no competing interests.

### Author Contributions

- Vasco Elbrecht conceived and designed the experiments, performed the experiments, analyzed the data, contributed reagents/materials/analysis tools, prepared figures and/or tables, authored or reviewed drafts of the paper, approved the final draft.

- Ecaterina Edith Vamos conceived and designed the experiments, performed the experiments, analyzed the data, authored or reviewed drafts of the paper, approved the final draft.
- Dirk Steinke conceived and designed the experiments, authored or reviewed drafts of the paper, approved the final draft.
- Florian Leese conceived and designed the experiments, authored or reviewed drafts of the paper, approved the final draft.

### Data Availability

Unprocessed raw sequence data are available from previous studies on the NCBI SRA archive. Single-species mock sample: SRR5295658 and SRR5295659 (*Vamos, Elbrecht & Leese, 2017*), monitoring samples: SRR4112287 (*Elbrecht et al., 2017*). The JAMP R package is available on GitHub (http://github.com/VascoElbrecht/JAMP) with the used R scripts (Script S1) and full haplotype tables (Table S1) available as Supplemental Dataset Files.

### Supplemental Information

Supplemental information for this article can be found online at http://dx.doi.org/10.7717/peerj.4644#supplemental-information.

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
