# Peer review of "Estimating intraspecific genetic diversity from community DNA metabarcoding data"

_PeerJ, doi:10.7717/peerj.4644_

## Round 0.1 · original submission · Major Revisions

· Academic Editor

Major Revisions

Based on the three reviews, I would ask you to pay particular attention to the comments of reviewers regarding the level of detail of your methodology (in the main text) and also the comments of reviewer 3 regarding the extensive supplementary material. As she indicates, if this material is not well-organised, it cannot fulfill its key function - please ensure that all material in this section is clearly explained and avoids unnecessary redundancies. Also, please pay particular attention to the comment of Reviewer 1 regarding the ability of CO1 variation to represent genetic variation - CO1 sequences provide information on marker variation, and their use as a broader proxy would require substantial supporting evidence which you do not provide. This said, I found this to be a very interesting paper, and I urge you to address all the reviewer comments by point, as this will be necessary to secure its successful publication.

Reviewer 1 ·

Basic reporting

Clear English yes.
Relevant literature references provided.
Code shared, good figures.
Objective needs to be clarified, see comments in the attachment.

Experimental design

Original primary research within scope, yes.
Research question should be re-framed, see attached comments.
Good preliminary investigation, see attached comments.
Wet lab methods not described in sufficient detail.
More clarity needed for bioinformatics methods, see attached comments.

Validity of the findings

Title and abstract should be edited to reflect the discussion which provides a more thoughtful and balanced perspective for using partial CO1 sequences for assessing intraspecific genetic diversity. See attached comments.

Additional comments

This paper is nice but would benefit from the addition of method details. This paper could be a particularly timely contribution to the literature if the objective is re-framed as looking at the increased resolution provided by ESVs compared with OTUs.

Annotated reviews are not available for download in order to protect the identity of reviewers who chose to remain anonymous.

·

Basic reporting

no comment

Experimental design

no comment

Validity of the findings

no comment

Additional comments

This is timely and useful research. I'm intrigued by it's potential and I look forward to trying this haplotype analysis in my own work.

I have one main question/concern: For the multi-species samples, were they IDed morphologically prior to metabarcoding? and if so, were the OTUs found in line with what had been found based on morphology?

My only other comments are minor grammatical changes:
Line 16: Change 'loosing' to 'losing'
Line 18: Hyphenate 'abundance based'
Line 33-34: Change 'to capture' to 'of capturing'
Line 49-50: There is a strange break before the period. Perhaps an improper line break/enter?
Line 53: Change 'milestone' to 'leap'
Line 57: Place a comma after 'management'
Line 74: Move 'collected' to between 'we' & 'multi-species'
Line 83: I am not sure why there's a '(1)'
Line 100: Change 'into' to 'in'
Line 114: Place commas before and after 'in fact'
Line 180: Hyphenate 'metabarcoding based'

Reviewer 3 ·

Basic reporting

The language quality is good, except in the M&M section where some typos and unprecisions were present.
L93 First mention of expected error (ee) should be written out.
L70 hyphen between metabarcoding and based
L74 “Subseqeuntly, we multi-species metabarcoding data…” might be a word missing in this sentence.
L88 brackets are in odd number.
L95, L98, L102. Different names for the same thing, respectively “monitoring bulk samples”, “bulk samples” and “monitoring samples”. Please pick one and use it throughout the MS.
L97 is it dereplication or deduplication?
L102 multi instead of Mulit
L100-102. What was the reference chosen for the Finnish multi species samples mapping?. Also with Vsearch or another mapper (Usearch?)

The background information provided in the introduction and the discussion of the results are appropriate and helpful for placing the present study into context.

Experimental design

This regards self containment and design both: to be able to reach a broader audience, this paper lacks in clarity and it is difficult for somebody new to the analysis to fully understand the steps taken. It helps to read the documentation available on Github for the JAMP package, but I am critical of papers where you actually have to read two other papers and a few Github commits to be able to follow. I am also puzzled by the Github readme of JAMP, which says: “At this point it is intended for internal @leeselab use ONLY”. Isn’t it a bit contradictory?

I have concerns about the material and methods section, which is essential in this paper since it is mostly a methodological one. One of the key criteria at PeerJ is a good description of the experimental design. While the choice of study sites/datasets and the various steps taken to test the several hypotheses are appropriate, the reporting isn’t.
I would recommend revisions, particularly in the very abundant supplementary material, to make it more understandable and eventually ensure a better dissemination. I did look at everything, and while I appreciate the authors’ openness, it is not usable as of now. In particular, it is not clear at first glance what is used as an input for what step. Also, sometimes I have the feeling several versions of the same script were uploaded. An example is denoise, I found denoise v6 and denoise v7. Which one is the appropriate one? Are they really different?
I am aware it is somewhat unusual to put so much emphasis on suppmats, but in this case, the work product is a software package, where documentation is essential.

I woudl like to make clear at this point that I am sure the authors did well and I do not doubt the results at all, only the amount/reporting of details provided.

Validity of the findings

The authors also honestly and clearly identify the limitations of their study or more generally of the approach itself. Furthermore, they point out how an approach of this kind could help identify interesting study sites/populations.

Additional comments

Elbrecht et al present a tool to mine the wealth of data „hidden“ in metabarcoding approaches. These are very widely used to describe community’s composition or subsets of these communities, and are based (most of times) on sequences of the cytochrome oxidase subunit I when analysing animal DNA. Elbrecht et al want to show that the intraspecific diversity can be described as well in a reliable manner, thus allowing going one step further in monitoring. I commend the initiative and think this study is very interesting from a personal point of view, and would be useful to a broader community.
The issues with the M&M section need to be addressed.

Suggestion for Figures:
I would increase the contrast in Fig1. A, the grey and blue symbols are very light. Furthermore, the extensive use of green and red make Panels A and C unsuitable for color-blind readers. Please consider using magenta and blue instead in Panel C. Same goes for the haplotype maps

---

## Round 0.2 · accepted · Accept

· Academic Editor

Accept

I am content that you have addressed all of the substantive comments of the reviewers, and the paper is in good shape for publication - thanks for your diligence in responding to all comments in a clear and concise manner.

#